# Did Affiliate Stigma Predict Affective and Behavioral Outcomes in Caregivers and Their Children with Attention-Deficit/Hyperactivity Disorder?

**DOI:** 10.3390/ijerph18147532

**Published:** 2021-07-15

**Authors:** Chih-Cheng Chang, Yu-Min Chen, Ray C. Hsiao, Wen-Jiun Chou, Cheng-Fang Yen

**Affiliations:** 1Department of Psychiatry, Chi Mei Medical Center, Tainan 70246, Taiwan; rabiata@mail.chimei.org.tw; 2Department of Health Psychology, College of Health Sciences, Chang Jung Christian University, Tainan 71101, Taiwan; 3Department of Psychiatry, Kaohsiung Medical University Hospital, Kaohsiung 80708, Taiwan; bluepooh79@msn.com; 4Department of Psychiatry, School of Medicine, and Graduate Institute of Medicine, Kaohsiung Medical University, Kaohsiung 80708, Taiwan; 5Department of Psychiatry and Behavioral Sciences, University of Washington School of Medicine, Seattle, WA 98195, USA; rhsiao@u.washington.edu; 6Department of Psychiatry, Children’s Hospital and Regional Medical Center, Seattle, WA 98105, USA; 7College of Medicine, Chang Gung University, Taoyuan 33302, Taiwan; 8Department of Child and Adolescent Psychiatry, Chang Gung Memorial Hospital, Kaohsiung Medical Center, Kaohsiung 83301, Taiwan

**Keywords:** attention-deficit/hyperactivity disorder, affiliate stigma, depression, behavioral problems

## Abstract

The aim of this follow-up study was to examine the predictive values of caregivers’ affiliate stigma at baseline for depression in caregivers and internalizing and externalizing symptoms in children with attention-deficit/hyperactivity disorder (ADHD) 1 year later. The Study on Affiliate Stigma in Caregivers of Children with ADHD surveyed the levels of affiliate stigma and depression in 400 caregivers and the behavioral problems of their children with ADHD. The levels of the caregivers’ depression and children’s behavioral problems were assessed 1 year later. The associations of caregivers’ affiliate stigma at baseline with depression in caregivers and internalizing and externalizing symptoms in children with ADHD at follow-up were examined using stepwise multiple regression. The results indicated that before caregivers’ depression and children’s behavioral problems at baseline were controlled, caregivers’ affiliate stigma at baseline positively predicted caregivers’ depression and all children’s behavioral problems. After caregivers’ depression and children’s behavioral problems at baseline were controlled, caregivers’ affiliate stigma at baseline still positively predicted children’s affective and somatic problems. Parenting training and cognitive behavioral therapy should be provided to caregivers with intense affiliate stigma to prevent emotional problems and difficulties in managing their children’s behavioral problems.

## 1. Introduction

### 1.1. Affiliate Stigma in Caregivers of Children with Attention-Deficit/Hyperactivity Disorder

Caregivers of children with neurodevelopmental disorders are prone to courtesy stigma [1] and affiliate stigma [2] when managing their children’s behavioral and emotional problems [3]. Courtesy stigma occurs when caregivers are mocked or blamed for their children’s illnesses or difficulties [2]. Consequently, caregivers may internalize courtesy stigma and develop negative attitudes toward themselves as part of a phenomenon called “affiliate stigma” [2]. Affiliate stigma may cause caregivers to feel desperate and helpless about their association with children with neurodevelopmental disorders and further compromise the caregivers’ mental health [4]. Affiliate stigma may also cause caregivers to withdraw from social interactions with supportive sources and medical services and even distance themselves from their children with neurodevelopmental disorders [5].

Attention-deficit/hyperactivity disorder (ADHD) is the most prevalent neurodevelopmental disorder, and it affects 4–10% of school-aged children [6]. However, affiliate stigma in caregivers of children with ADHD has been examined less than the stigma experienced by caregivers of children with other neurodevelopmental disorders has, such as intellectual disability and autism spectrum disorder [2,3]. Research has revealed that affiliate stigma is negatively associated with the levels of caregivers’ agreement with their children’s ADHD diagnosis, the necessity of treatment for the child’s ADHD, and whether biological etiologies explain the child’s ADHD diagnosis [7]. Evaluating and reducing affiliate stigma in caregivers of children with ADHD will help mental health professionals provide early diagnosis and essential treatment to children and their caregivers.

### 1.2. Relationship between Affiliate Stigma and Depression in Caregivers of Children with ADHD

Several studies have determined that caregivers of children with ADHD have high parenting stress [8,9] and care burdens [10,11,12] and that consequently, a large proportion of caregivers of children with ADHD develop depression [13]. Depression further negatively affects caregivers’ self-efficacy [14] and quality of interaction with their children [15,16]. Determining the predictors of depression is essential to developing prevention programs for caregivers of children with ADHD. A cross-sectional study demonstrated a positive association between affiliate stigma and distress in mothers of male children with ADHD [17]. A 2021 study also supported the cross-sectional relationship between affiliate stigma and depression level in caregivers of children with ADHD [18]. However, whether affiliate stigma can longitudinally predict the level of depression potentially faced by caregivers of children with ADHD warrants further study.

### 1.3. Relationship of Caregivers’ Affiliate Stigma with Internalizing and Externalizing Symptoms in Children with ADHD

ADHD is often accompanied by affective and behavioral problems [19] that are not included in the ADHD criteria listed in the *Diagnostic and Statistical Manual of Mental Disorders, Fifth Edition* (DSM-5) [20]. Multiple studies have evidenced that people with ADHD have more comorbid internalizing (e.g., depressive, anxious, and somatic symptoms) and externalizing symptoms (e.g., oppositional defiance and conduct problems) than do people without ADHD [21,22,23,24,25,26]. Comorbid internalizing and externalizing symptoms not only worsen functional outcomes during childhood [27] but also predict ADHD persistence from childhood into adulthood and adverse outcomes in adulthood among children with ADHD [28]. Therefore, comorbid internalizing and externalizing symptoms in children with ADHD warrant routine assessment and early intervention.

Caregivers’ affiliate stigma may interfere with caregivers’ parenting behaviors and affective interaction with children with ADHD [29]. Moreover, children may lack the caregiver support and discipline that they require to develop strategies to regulate their emotions and behaviors. Therefore, affiliate stigma may predict internalizing and externalizing symptoms in children with ADHD. Cross-sectional studies have found a significant positive association between caregivers’ affiliate stigma and the internalizing [18] and externalizing symptoms [29] of children with ADHD. Further follow-up studies are warranted to examine the ability of affiliate stigma to predict comorbid internalizing and externalizing symptoms in children with ADHD.

### 1.4. Aims of This Study

The Study on Affiliate Stigma in Caregivers of Children with ADHD, which was conducted between June 2019 and April 2020, surveyed the levels of affiliate stigma and depression in 400 caregivers and the behavioral problems of their children with ADHD. The present study followed up the caregivers and children with ADHD 1 year later and resurveyed the caregivers’ depression and children’s behavioral problems. This 1-year follow-up study had two aims. First, we aimed to examine the predictive value of caregivers’ affiliate stigma at baseline for caregivers’ depression 1 year later. Second, we aimed to examine the predictive value of caregivers’ affiliate stigma at baseline for internalizing and externalizing symptoms in children with ADHD 1 year later. Accordingly, this study hypothesized the following:

**Hypothesis** **1 (H1).**
*C*
*aregivers’ affiliate stigma at baseline predicts positively caregivers’ depression 1 year later.*


**Hypothesis** **2 (H2).***C**aregivers’ affiliate stigma at baseline predicts internalizing and externalizing symptoms in children with ADHD 1 year later*.

## 2. Methods

### 2.1. Participants and Procedures

The procedure of recruiting participants at baseline for the Study on Affiliate Stigma in Caregivers of Children with ADHD in Taiwan has been described elsewhere [18]. In brief, 400 caregivers of children aged 6–18 years who had been diagnosed as having ADHD according to DSM-5 criteria [20] were recruited from the child and adolescent psychiatric outpatient clinics of two medical centers in Kaohsiung, Taiwan. Children who had an intellectual disability or autism spectrum disorder with difficulties in communication had been excluded. Caregivers who had an intellectual disability, schizophrenia, bipolar disorder, or any cognitive deficits that resulted in significant communication difficulties had been also excluded. The levels of affiliate stigma and depression in caregivers and their children’s ADHD, affective, anxiety, somatic, oppositional defiant, and conduct problems were assessed. One year later, these caregivers were contacted at the outpatient clinics again and invited to complete follow-up assessments of their depression and children’s behavioral problems on the same research questionnaires. 

The institutional review boards of Kaohsiung Chang Gung Memorial Hospital (approval number: 202000605A3; date of approval: 15 May 2020) and Kaohsiung Medical University Hospital (approval number: KMUHIRB-SV(I)-20190130 and KMUHIRB-E(I)-20200111; date of approval: 7 May 2019 and 1 June 2020) approved the study. All participants provided written informed consent before receiving the assessment.

### 2.2. Measures

#### 2.2.1. Predictor: Affiliate Stigma

The 22-item Affiliate Stigma Scale was used to assess caregivers’ internalized stigma toward their children’s ADHD at baseline [5,30]. Each item was rated on a 4-point Likert scale from 1 (*strongly disagree*) to 4 (*strongly agree*). A higher total score represented a higher level of affiliate stigma. Cronbach’s α for the scale in the present study was 0.95.

#### 2.2.2. Outcomes: Caregivers’ Depression

The 20-item Mandarin Chinese version of the Center for Epidemiological Studies Depression Scale (CES-D) was applied to assess the severity of self-reported depressive symptoms in caregivers at baseline and at follow-up [31,32]. Caregivers were asked how often they experienced each depressive symptom in the preceding month. The response categories, rated from 0 to 3 and classified according to frequency in the last month, were as follows: “rarely or none of the time” (0; frequency: less than 1 day), “some or a little of the time” (1; frequency: l–2 days), “occasionally or a moderate amount of the time” (2; frequency: 3–4 days), and “most or all of the time” (3; frequency: 5–7 days). A higher total score indicated more severe depression. Cronbach’s α for the scale in the present study was 0.88.

#### 2.2.3. Outcomes: Children’s Behavioral Problems

The caregiver-reported Chinese version of the Child Behavior Checklist for Ages 6–18 (CBCL/6-18) was used to measure children’s behavioral problems at baseline and at follow-up [33,34,35]. We used the recommended T-score transformations of raw behavior scores, which were adjusted for age and sex differences in behavior that had been found in normative samples. We used the domains of affective, anxiety, somatic, oppositional defiant, and conduct problems in our analysis.

#### 2.2.4. Covariates: Demographic Characteristics and ADHD Symptoms

Caregivers’ sex, age, marital status, and education level, as well as children’s sex and age, were collected at baseline. ADHD symptoms on CBCL/6-18 at baseline was used as a covariate because children in this study were recruited from the clinical units and received pharmacological or psychological treatment for their ADHD.

### 2.3. Statistical Analysis

Data analysis was performed using SPSS 24.0 (SPSS, Chicago, IL, USA). Sociodemographics, caregivers’ affiliate stigma and depression, and children’s internalizing and externalizing problems at baseline were compared between caregivers who completed the follow-up assessment and those who did not using chi-square and *t* tests. Caregivers’ demographic characteristics, affiliate stigma, and depressive symptoms, as well as children’s demographic characteristics and behavioral problems, are presented as percentages and means and standard deviations. Correlations between the continuous variables were examined using Pearson’s correlation. The value of coefficient ranging between 0.5 and 1 was classified to have a large and significant strength of association [36]. Caregivers’ depression and children’s behavioral symptoms between at baseline and at follow-up were compared using paired *t*-tests.

The ability of caregivers’ affiliate stigma at baseline (predictor) to predict caregivers’ depressive symptoms and children’s behavioral problems at follow-up (outcomes) were examined using two forward stepwise multiple regression models. The first model (Model I) controlled for the effects of caregivers’ and children’s demographic characteristics and ADHD symptoms at baseline (covariates); the second model (Model II) further controlled for the effects of caregivers’ depression and children’s behavioral problems at baseline, in addition to their demographic characteristics. As multiple comparisons were made, a two-tailed *p* value < 0.008 (0.05/6) was set to indicate statistical significance.

## 3. Results

In total, 382 caregivers (95.5%, 308 women and 74 men) of children with ADHD completed a 1-year follow-up assessment. No difference in caregivers’ sex (χ^2^ = 0.766, *p* = 0.381), age (*t* = 0.475, *p* = 0.635), marital status (χ^2^ = 1.998, *p* = 0.158), education level (*t* = −1.325, *p* = 0.186), affiliate stigma (*t* = −1.250, *p* = 0.212), or depression (*t* = −0.271, *p* = 0.787), nor any differences in children’s sex (χ^2^ = 0.845, *p* = 0.358), age (*t* =−0.903, *p* = 0.367), and behavioral problems (*t* = −1.121– −0.672, *p* = 0.263–0.502) at baseline, were found between caregivers who completed the follow-up assessment and those who did not. Table 1 presents the baseline characteristics of the 382 caregivers and children with ADHD.

Table 2 presents the correlations between affiliate stigma and ADHD symptoms at baseline, caregivers’ depression levels, and children’s internalizing and externalizing symptoms at baseline and at follow-up, which were all examined using Pearson’s correlation. At baseline, the large strength of association was noted between caregivers’ affiliate stigma and depression, between children’s affective and anxious problems, between children’s affective and conduct problems, and among children’s ADHD symptoms, ODD and conduct problems. At follow-up, the large strength of association was noted between caregivers’ depression and children’s affective problems, between children’s affective and anxious problems, between children’s affective and ODD problems, and between children’s ODD and conduct problems. Caregivers’ depression at baseline and at follow-up correlated with each other. Children’s internal and external symptoms at baseline and at follow-up also correlated with each other except for somatic problems. The correlation between affiliate stigma and caregivers’ depression from 0.508 at baseline decreased to 0.407 at follow-up. Although the correlations of affiliate stigma with children’s behavioral problems were not the same compared between at baseline and at follow-up, the correlations did not reach a large and significant level.

The results of paired *t* tests comparing caregivers’ depression and children’s behavioral symptoms between at baseline and at follow-up demonstrated that children’s affective (*p* < 0.001) and conduct problems (*p* = 0.005) decreased significantly between at baseline and at follow-up. Children’s anxiety (*p* = 0.020) and ODD problems (*p* = 0.021) tended to decrease but did not reach the level of significance. Caregivers’ depression (*p* = 0.111) and children’s somatic symptoms (*p* = 0.263) did not change significantly.

Table 3 presents the results of the stepwise multiple regression analysis that examined the associations between caregivers’ affiliate stigma at baseline and caregivers’ depression and children’s behavioral problems at follow-up. The results demonstrated that, after the effects of demographic characteristics and ADHD symptoms were controlled, caregivers’ affiliate stigma at baseline was positively associated with caregivers’ depression and all children’s behavioral problems at follow-up (Model I). The association of affiliate stigma with caregivers’ depression was the strongest (B = 0.318) compared with the associations of affiliate stigma with children’s behavioral problems (B = 0.113~0.260). After caregivers’ depression and children’s behavioral problems at baseline were added to the stepwise multiple regression analysis, caregivers’ affiliate stigma at baseline was still positively associated with children’s affective and somatic problems at follow-up, though the magnitude of the regression coefficients decreased significantly (B of affective problems: decreasing from 0.260 to 0.119; B of somatic problems: decreasing from 0.113 to 0.085) (Model II).

## 4. Discussion

This study is one of the first to examine the predictive ability of affiliate stigma for caregivers’ depression and children’s behavioral problems at 1-year follow-up. The present study demonstrated that before caregivers’ depression and children’s behavioral problems and ADHD symptoms at baseline were controlled, caregivers’ affiliate stigma at baseline positively predicted caregivers’ depression and all children’s behavioral problems. After caregivers’ depression and children’s behavioral problems at baseline were controlled, caregivers’ affiliate stigma at baseline still positively predicted children’s affective and somatic problems.

### 4.1. Affiliate Stigma’s Ability to Predict Depression in Caregivers of Children with ADHD

The present study found that affiliate stigma at baseline correlated positively with depression at baseline and at follow-up in caregivers of children with ADHD. However, the predictive ability of affiliate stigma at baseline for depression at follow-up became nonsignificant after depression at baseline was controlled. Affiliate stigma, which occurs when caregivers internalize negative attitudes toward themselves, may exacerbate caregivers’ depression in several ways. First, affiliate stigma may demoralize caregivers by making them feel ashamed. Second, caregivers with intense affiliate stigma may withdraw from social interactions [5,37] and consequently lack social support in child-reading and in their emotions. Third, depression may further compromise caregiver–child interaction and intensify conflict. Moreover, both affiliate stigma and depression may result from caregivers’ negative and self-blaming cognition [29,38]. Thus, affiliate stigma and depression may form a vicious cycle and reinforce each other. The results of the present study suggest that affiliate stigma indicates depression in caregivers of children with ADHD. Intervention programs that enhance caregivers’ mental health are required to reduce affiliate stigma.

### 4.2. Affiliate Stigma’s Ability to Predict Internalizing and Externalizing Symptoms in Children with ADHD

The present study shows that, before children’s behavioral problems at baseline were controlled, caregivers’ affiliate stigma at baseline positively predicted all behavioral problems in the children at 1-year follow-up. After children’s behavioral problems at baseline were controlled, caregivers’ affiliate stigma at baseline still positively predicted children’s affective and somatic problems at follow-up. Our findings that adverse consequences of affiliate stigma may extend beyond negative effects on caregivers who internalize negative attitudes from others to their relationship with children who have engendered the stigma for caregivers are consistent with those of another study [29]. The predictive effects of affiliate stigma on children’s internalizing problems vary. First, affiliate stigma may lead caregivers to respond more critically when their children exhibit ADHD symptoms [29]; such criticism may not only provoke further conflicts between children and caregivers but also reduce children’s confidence. Second, caregivers with affiliate stigma may distance themselves from their children with ADHD to avoid feeling shame [1]; therefore, children may not receive sufficient psychological support to properly develop emotional regulation. Third, caregivers with strong affiliate stigma tend to attribute their children’s ADHD to nonbiological etiologies [39] and employ intervention programs that lack credibility [40]. Children’s ADHD symptoms may not be treated adequately and consequently impair multiple dimensions of children’s functioning.

### 4.3. Implications

The results of this study address the importance of mitigating affiliate stigma in caregivers of children with ADHD. According to Ecological Systems Theory [41], the work of mitigating affiliate stigma should be started from the individual and environmental levels simultaneously. Regarding the individual level, because knowledge is essential to caregivers’ positive attitudes toward children and themselves, behavioral training programs for caregivers should increase their knowledge regarding the etiologies, illness courses, and treatment models of ADHD [42,43]. Training programs should also increase caregivers’ resistance to other people’s negative attitudes [29,44]. Regarding the environmental level, enhancing social support for caregivers may increase their positive attitudes toward themselves and reduce social isolation. Moreover, mitigating public stigma toward ADHD is also essential to decreasing caregivers’ affiliate stigma.

The present study revealed the predictive value of affiliate stigma for depression in caregivers and internalizing and externalizing symptoms in children. Mental health professionals should also routinely survey caregivers’ affiliate stigma and provide cognitive behavioral therapy in combination with behavioral parental training to help caregivers with intense affiliate stigma reduce their depression and improve their ability to manage their children’s behavioral problems.

### 4.4. Limitations

This study had several limitations. First, the present study did not survey the level of affiliate stigma at follow-up and could not determine whether the changes in affiliate stigma influenced caregivers’ depression and children’s behavioral problems at follow-up. Second, the data collected in this study were provided by the caregivers. This single data source may have resulted in common method variance. Third, research has found that both paternal depression and maternal depression in the pre-pregnancy, perinatal and postnatal periods increase offspring’s ADHD risk [45], as well as that mothers’ depressive symptoms predicted negative biases in their reports of their child’s ADHD symptoms, general behavior problems, and their own negative parenting style [46]. We did not collect caregivers’ preexisting mental health problems before having children or having difficulties in managing their child’s ADHD problems; therefore, we could not determine whether caregivers’ preexisting mental health problems have additive effects on caregivers’ depression and children’s behavioral problems at follow-up.

## 5. Conclusions

This study found that affiliate stigma in caregivers of children with ADHD at baseline can predict depression in caregivers as well as the internalizing and externalizing problems of children with ADHD one year later. Based on the results of this study we suggested that behavioral training programs for caregivers should mitigate affiliate stigma by providing caregivers with psychoeducation to increase their knowledge of ADHD and strengthen their resistance to stigmas held by people around them. Meanwhile, cognitive behavioral therapy and behavioral parental training should be provided to caregivers with intense affiliate stigma to prevent emotional problems and difficulties in managing their children’s behavioral problems.

## Figures and Tables

**Table 1 ijerph-18-07532-t001:** Baseline and follow-up data of caregivers and children with ADHD (*N* = 382).

Variable	*n* (%)	Mean (SD)	Range
*Baseline*			
Caregivers			
Gender			
Female	308 (80.6)		
Male	74 (19.4)		
Age (years)		43.1 (7.0)	23–69
Education of year (years)		14.2 (3.2)	0–28
Marriage status			
Married or cohabited	307 (80.4)		
Separated or divorced	75 (19.6)		
Affiliate stigma		37.4 (11.0)	22–71
Depression		13.8 (9.6)	0–45
Children			
Gender			
Girl	76 (19.9)		
Boy	306 (80.1)		
Age (years)		10.9 (3.2)	6–18
ADHD symptoms		64.1 (7.7)	50–80
Affective problems		64.1 (8.5)	50–90
Anxiety problems		59.9 (7.8)	50–77
Somatic problems		55.8 (8.7)	50–100
Oppositional defiant problems		61.0 (8.5)	50–80
Conduct problems		59.8 (8.6)	50–88
*Follow-up*			
Caregivers			
Depression		14.4 (9.9)	0–55
Children			
Affective problems		62.3 (8.7)	50–95
Anxiety problems		59.0 (8.3)	50–78
Somatic problems		55.3 (7.5)	50–97
Oppositional defiant problems		60.1 (7.1)	50–80
Conduct problems		58.9 (8.0)	50–81

ADHD: attention deficit hyperactivity disorder; SD: standard deviation.

**Table 2 ijerph-18-07532-t002:** Correlations among affiliate stigma and attention deficit hyperactivity disorder symptoms at baseline, caregivers’ depression, and children’s internalizing and externalizing symptoms at baseline and at follow-up: Pearson’s correlation.

		1	2	3	4	5	6	7	8	9	10	11	12	13	14
At baseline	1. Affiliate stigma	-													
	2. ADHD symptoms	0.254	-												
	3. Depression	**0.508**	0.227	-											
	4. Affective problems	0.350	0.455	0.382	-										
	5. Anxious problems	0.354	0.348	0.324	**0.615**	-									
	6. Somatic problems	0.111	0.137	0.164	0.404	0.277	-								
	7. ODD problems	0.288	**0.608**	0.290	0.452	0.329	0.162	-							
	8. Conduct problems	0.349	**0.618**	0.279	**0.534**	0.350	0.281	**0.707**	-						
At follow-up	9. Depression	0.407	0.279	**0.692**	0.379	0.323	0.192	0.201	0.265	-					
	10. Affective problems	0.330	0.318	0.324	**0.559**	0.461	0.266	0.274	0.314	**0.520**	-				
	11. Anxious problems	0.262	0.210	0.215	0.413	**0.576**	0.274	0.173	0.227	0.374	**0.645**	-			
	12. Somatic problems	0.166	0.187	0.129	0.300	0.291	0.443	0.152	0.207	0.239	0.428	0.400	-		
	13. ODD problems	0.248	0.393	0.219	0.333	0.257	0.178	**0.582**	0.497	0.354	**0.** **515**	0.409	0.189	-	
	14. Conduct problems	0.297	0.451	0.199	0.353	0.261	0.169	0.459	**0.688**	0.358	0.499	0.385	0.229	**0.701**	-

ODD: oppositional defiant disorder; values in bold if >0.5.

**Table 3 ijerph-18-07532-t003:** Associations of caregivers’ affiliate stigma at baseline with caregivers’ depression and children’s behavioral problems at follow-up: forward stepwise multiple regression analysis ^a^.

	Caregivers’ Depression at Follow-Up		Children’s Behavioral Problems at Follow-Up									
			Affective Problems		Anxiety Problems		Somatic Problems		Oppositional Defiant Problems		Conduct Problems	
	Model I	Model II	Model I	Model II	Model I	Model II	Model I	Model II	Model I	Model II	Model I	Model II
Baseline	B (SE)	B (SE)	B (SE)	B (SE)	B (SE)	B (SE)	B (SE)	B (SE)	B (SE)	B (SE)	B (SE)	B (SE)
caregivers												
Affiliate stigma	0.318 (0.042) ***	0.048 (0.038)	0.260 (0.038) ***	0.119 (0.035) **	0.197 (0.037) ***		0.113 (0.034) **	0.085 (0.031) **	0.187 (0.036) ***	0.074 (0.032) *	0.216 (0.035) ***	0.058 (0.028) *
Depression		0.651 (0.044) ***										
Children												
Affective problems				0.514 (0.046) ***								
Anxiety problems						0.584 (0.047) ***						
Somatic problems								0.522 (0.042) ***				
Oppositional defiant problems										0.522 (0.042) ***		
Conduct problems												0.594 (0.036) ***

^a^: Controlling for the effects of caregiver’s sex, age, education, and marital status and children’s sex, age, and ADHD symptoms; **: p* < 0.05; ***: p* < 0.01; ****: p* < 0.001.

## Data Availability

The data will be available upon reasonable request to the corresponding authors.

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
