# Peer review of "Did Affiliate Stigma Predict Affective and Behavioral Outcomes in Caregivers and Their Children with Attention-Deficit/Hyperactivity Disorder?"

_ijerph, 2021, doi:10.3390/ijerph18147532_

Round 1

Reviewer 1 Report

Title Page: Personal informations msut be editted. All characters should have the same type and size.

Introduction: A good introduction on the problematic is presented. It is of course focused to investigat the link between caregiver affiliate stigma and mental illness. It would be though interesting to have some background about the earlier mental health of those caregivers, before they have children or before their children got the ADHD diagnosis. It could be that they were already proned to such distress?

Methods: Lines 145-146: was the affiliate stigma questionnair eonly applied once at baseline? What about the follow-up?

Statistical analysis: please give the correlations cut-off references. What does it mean to have a correlation of 0.4, for example?

Which variable selection method was used for the stepwise regression? Which statistical software?

Results: Group comparisons were performed but the statistical testes (Chi-square, t-Test) were not described in Methods.

Table 2: It would be more interesting to comment the correlations coefficient than their statistical significance. Please add some cut-off references and discuss on that.

A discussion regarding the magnitude of the regression coefficients and their associations would be desirable. What does the coefficients means? What the values tell the reader?

Author Response

Comment 1

Title Page: Personal informations msut be editted. All characters should have the same type and size.

Response

Thank you for your reminding. We unified the type and size of characters in authors’ information. Please refer to line 7 and 11.

Comment 2

Introduction: It would be though interesting to have some background about the earlier mental health of those caregivers, before they have children or before their children got the ADHD diagnosis. It could be that they were already proned to such distress?

Response

Thank you for your suggestion. We agreed that caregivers’ earlier mental health may influence their proneness to distress and interaction with children with ADHD. However, we did not collect caregivers’ earlier mental health and could not determine their effects. We listed it as one of the limitations in this study as below. Please refer to line 353-363.

“Third, research has found that both paternal depression and maternal depression in the pre-pregnancy, perinatal and postnatal periods increase offspring’s ADHD risk [45], as well as that mothers' depressive symptoms predicted negative biases in their reports of their child's ADHD symptoms, general behavior problems, and their own negative parenting style [46]. We did not collect caregivers’ preexisting mental health problems before having children or having difficulties in managing their child’s ADHD problems; therefore, we could not determine whether caregivers’ preexisting mental health problems have additive effects on caregivers’ depression and children’s behavioral problems at follow-up.”

Comment 3

Methods: Lines 145-146: was the affiliate stigma questionnair eonly applied once at baseline? What about the follow-up?

Response

Thank you for your comment. In our original study idea, caregiver’s affiliate stigma was considered as the predictor, and the follow-up study aimed to examined its predictive values for caregivers’ depression and children’s behavioral problems one year late. Therefore, we did not reassess caregivers’ affiliate stigma at follow-up. However, it limited the possibility to determine whether the changes in affiliate stigma influenced caregivers’ depression and children’s behavioral problems at follow-up. We have listed it as one of the limitations in the original manuscript. Please refer to line 348-351.

Comment 4

Statistical analysis: please give the correlations cut-off references. What does it mean to have a correlation of 0.4, for example?

Response

Thank you for your suggestion. In the revised manuscript we used 0.5 as the cutoff reference. We added it into the section of Statistical Analysis as below. Please refer to line 193-195.

“The value of coefficient ranging between 0.5 and 1 was classified to have a large and significant strength of association [36].

Comment 5

Which variable selection method was used for the stepwise regression? Which statistical software?

Response

  1. We used “forward” stepwise multiple We added it into the revised manuscript. Please refer to line 199 and 270.
  2. We used SPSS 24.0. We added it into the revised manuscript. Please refer to line 185.

“Data analysis was performed using SPSS 24.0 (SPSS, Chicago, IL, USA).”

Comment 6

Results: Group comparisons were performed but the statistical testes (Chi-square, t-Test) were not described in Methods.

Response

Thank you for your reminding. We added it into the revised manuscript as below. Please refer to line 186-189.

Sociodemographics, caregivers’ affiliate stigma and depression, and children’s internalizing and externalizing problems at baseline were compared between caregivers who completed the followup assessment and those who did not using chi-square and t tests.

Comment 7

Table 2: It would be more interesting to comment the correlations coefficient than their statistical significance. Please add some cut-off references and discuss on that.

Response

Thank you for your suggestion. We added the cut-off reference for the correlations coefficient and interpret the results of correlation test into the revised manuscript as below.

“The value of coefficient ranging between 0.5 and 1 was classified to have a large and significant strength of association [36].” Please refer to line 193-195.

At baseline, the large strength of association was noted between caregivers’ affiliate stigma and depression, between children’s affective and anxious problems, between children’s affective and conduct problems, and among children’s ADHD symptoms, ODD and conduct problems. At follow-up, the large strength of association was noted between caregivers’ depression and children’s affective problems, between children’s affective and anxious problems, between children’s affective and ODD problems, and between children’s ODD and conduct problems. Caregivers’ depression at baseline and at follow-up correlated with each other. Children’s internal and external symptoms at baseline and at follow-up also correlated with each other except for somatic problems. The correlation between affiliate stigma and caregivers’ depression from 0.508 at baseline decrease to 0.407 at follow-up. Although the correlations of affiliate stigma with children’s behavioral problems were not the same compared between at baseline and at follow-up, the correlations did not reach a large and significant level.  Please refer to line 224-239.

Comment 8

A discussion regarding the magnitude of the regression coefficients and their associations would be desirable. What does the coefficients means? What the values tell the reader?

Response

Thank you for your suggestion. We added more interpretations for the magnitude of the regression coefficients as below. Please refer to line 258-267.

“…(Model I). The association of affiliate stigma with caregivers’ depression was the strongest (B = 0.318) compared with the associations of affiliate stigma with children’s behavioral problems (B = 0.113~0.260). After caregivers’ depression and children’s behavioral problems at baseline were added to the stepwise multiple regression analysis, caregivers’ affiliate stigma at baseline was still positively associated with children’s affective and somatic problems at follow-up, though the magnitude of the regression coefficients decreased significantly (B of affective problems: decreasing from 0.260 to 0.119; B of somatic problems: decreasing from 0.113 to 0.085) (Model II).”

Reviewer 2 Report

Please define the purpose of the research and relate it to the topic of the article.
Provide research questions.
Enter the independent and dependent variables.
Please provide the research hypothesis.
Please specify the inclusion and exclusion criteria from the research.
Conclusions should be closely related to the purpose of the research and research questions.

Author Response

Comment 1

Please define the purpose of the research and relate it to the topic of the article.
Response

Thank you for your suggestion. We listed the purpose of the research as below in Introduction.

“Determining the predictors of depression is essential to developing prevention programs for caregivers of children with ADHD. A cross-sectional study demonstrated a positive association between affiliate stigma and distress in mothers of male children with ADHD [17]. A 2021 study also supported the cross-sectional relationship between affiliate stigma and depression level in caregivers of children with ADHD [18]. However, whether affiliate stigma can longitudinally predict the level of depression potentially faced by caregivers of children with ADHD warrants further study.” Please refer to line 79-87.

“Cross-sectional studies have found a significant positive association between caregivers’ affiliate stigma and the internalizing [18] and externalizing symptoms [29] of children with ADHD. Further follow-up studies are warranted to examine the ability of affiliate stigma to predict comorbid internalizing and externalizing symptoms in children with ADHD.” Please refer to line 107-111.

Comment 2

Provide research questions.
Response

Thank you for your suggestion. We revised the research questions as below. Please refer to line 118-123.

This 1-year follow-up study had two aims. First, we aimed to examine the predictive value of caregivers’ affiliate stigma at baseline for caregivers’ depression 1 year later. Second, we aimed to examine the predictive value of caregivers’ affiliate stigma at baseline for internalizing and externalizing symptoms in children with ADHD 1 year later.

Comment 3

Enter the independent and dependent variables.
Response

We labelled the predictor, outcomes, and covariates as below in the revised manuscript.

“2.2.1. Predictor: Affiliate Stigma” Please refer to line 152.

“2.2.2. Outcomes: Caregivers’ Depression” Please refer to line 158.

“2.2.3. Outcomes: Children’s Behavioral Problems” Please refer to line 170.

“2.2.4. Covariates: Demographic Characteristics and ADHD Symptoms” Please refer to line 178.

“The ability of caregivers’ affiliate stigma at baseline (predictor) to predict caregivers’ depressive symptoms and children’s behavioral problems at follow-up (outcomes) were examined using two forward stepwise multiple regression models. The first model (Model I) controlled for the effects of caregivers’ and children’s demographic characteristics and ADHD symptoms at baseline (covariates)…”Please refer to line 197-202.

Comment 4

Please provide the research hypothesis.

Response

Thank you for your suggestion. We revised the research hypothesis as below. Please refer to line 123-127.

“Accordingly, this study hypothesized the following:

Hypothesis 1 (H1): Caregivers’ affiliate stigma at baseline predicts positively caregivers’ depression 1 year later.

Hypothesis 2 (H2): Caregivers’ affiliate stigma at baseline predicts internalizing and externalizing symptoms in children with ADHD 1 year later.”

Comment 5
Please specify the inclusion and exclusion criteria from the research.
Response

We added the inclusion and exclusion criteria of the he Study on Affiliate Stigma in Caregivers of Children with ADHD in Taiwan into the revised manuscript.

“Caregivers of children aged 6–18 years who had been diagnosed as having ADHD according to DSM-5 criteria [20] were recruited from the child and adolescent psychiatric outpatient clinics of two medical centers in Kaohsiung, Taiwan. Children who had an intellectual disability or autism spectrum disorder with difficulties in communication had been excluded. Caregivers who had an intellectual disability, schizophrenia, bipolar disorder, or any cognitive deficits that resulted in significant communication difficulties had been also excluded.” Please refer to line 132-139.

Comment 6

Conclusions should be closely related to the purpose of the research and research questions.

Response

We revised the content of Conclusion section as below. Please refer to line 365-374.

“This study found that affiliate stigma in caregivers of children with ADHD at baseline can predict depression in caregivers as well as the internalizing and externalizing problems of children with ADHD one year later. Based on the results of this study we suggested that behavioral training programs for caregivers should mitigate affiliate stigma by providing caregivers with psychoeducation to increase their knowledge of ADHD and strengthen their resistance to stigmas held by people around them. Meanwhile, cognitive behavioral therapy and behavioral parental training should be provided to caregivers with intense affiliate stigma to prevent emotional problems and difficulties in managing their children’s behavioral problems.”

Reviewer 3 Report

The article presented here focuses on the relationships between the mental health of parents, mainly mothers, and the symptomatology and mental health of their children, mainly boys, with ADHD. It assesses a large sample at two different points in time and relates the information obtained using different instruments. I believe that this article is a lot of work and is of interest.
Below I will point out some aspects of the article:
- Throughout the article I have missed the post-stigma assessment, and it is when we come to the limitations when it is pointed out that it has only been applied in the first assessment. I think this is an important limitation, but it should be reflected in the methodology. In this sense, it could be pointed out why this was done.
- Regarding the tables, I think that care should be taken that they are not cut on the page, I think that table 2 is a bit sloppy and it is not clear from where the following data are presented.
- In the results, I think that the ADHD symptomatology data should appear in the follow-up, the correlations appear but not the descriptive data.On the other hand, much emphasis is placed on the predictive value of parental stigma, but the results for pre and post symptomatology are quite similar: in children, symptomatology decreases in all areas except Behaviour Problems, and in parents, depressive symptomatology increases. It seems that the relationship of depressive symptomatology at follow-up is stronger than at the first assessment, as well as stigma.  It is possible that the long-term situation and the lack of support are affecting the families.
- The article talks about the difficulty that stigma and blaming of families can pose, and concludes by stressing that it is the parents who need treatment, psycho-educational or cognitive-behavioural, all of which can have an impact on blaming.  I believe that a systemic approach would improve the outlook on this situation and would de-blame the families.

Author Response

Comment 1

Throughout the article I have missed the post-stigma assessment, and it is when we come to the limitations when it is pointed out that it has only been applied in the first assessment. I think this is an important limitation, but it should be reflected in the methodology. In this sense, it could be pointed out why this was done.

Response

Thank you for your comment. In our original study idea, caregiver’s affiliate stigma was considered as the predictor, and the follow-up study aimed to examined its predictive values for caregivers’ depression and children’s behavioral problems one year late. Therefore, we did not reassess caregivers’ affiliate stigma at follow-up. However, it limited the possibility to determine whether the changes in affiliate stigma influenced caregivers’ depression and children’s behavioral problems at follow-up. In addition to list it as one of limitations of this study, we labelled the predictor, outcomes, and covariates as below in the revised manuscript to illustrate the original study idea.

“2.2.1. Predictor: Affiliate Stigma” Please refer to line 152.

“2.2.2. Outcomes: Caregivers’ Depression” Please refer to line 158.

“2.2.3. Outcomes: Children’s Behavioral Problems” Please refer to line 170.

“2.2.4. Covariates: Demographic Characteristics and ADHD Symptoms” Please refer to line 178.

Comment 2

Regarding the tables, I think that care should be taken that they are not cut on the page, I think that table 2 is a bit sloppy and it is not clear from where the following data are presented.

Response

Thank you for your comment. In the revised manuscript we used 0.5 as the cutoff reference for the Pearson’s correlation coefficient. Therefore, we revised the content of Table 2 and label the coefficients higher than 0.5 in bald. We also put the table in a page to make it easily to read. Please refer to line 248-251.

Comment 3
In the results, I think that the ADHD symptomatology data should appear in the follow-up, the correlations appear but not the descriptive data.

Response

Thank you for your reminding. Children’s ADHD symptoms at baseline were used as a covariate. We used it as a covariate because children in this study were recruited from the clinical units and received pharmacological or psychological treatment for their ADHD. We added the explanation as below into the revised manuscript. Please refer to line 178-183.

“2.2.4. Covariates: Demographic Characteristics and ADHD Symptoms

ADHD symptoms on CBCL/6-18 at baseline was used as a covariate because children in this study were recruited from the clinical units and received pharmacological or psychological treatment for their ADHD.”

Comment 4

On the other hand, much emphasis is placed on the predictive value of parental stigma, but the results for pre and post symptomatology are quite similar: in children, symptomatology decreases in all areas except Behaviour Problems, and in parents, depressive symptomatology increases. It seems that the relationship of depressive symptomatology at follow-up is stronger than at the first assessment, as well as stigma.  It is possible that the long-term situation and the lack of support are affecting the families.

Response

Thank you for your suggestion. We added the results of paired t tests comparing caregivers’ depression and children’s behavioral symptoms between at baseline and at follow-up as below. We also added interpretation for the correlations between affiliate stigma and caregivers’ depression and children’s behavioral problems as below.

“Caregivers’ depression and children’s behavioral symptoms between at baseline and at follow-up were compared using paired t tests.” Please refer to line 195-196.

“The results of paired t tests comparing caregivers’ depression and children’s behavioral symptoms between at baseline and at follow-up demonstrated that children’s affective (p < 0.001) and conduct problems (p = 0.005) decreased significantly between at baseline and at follow-up. Children’s anxiety (p = 0.020) and ODD problems (p = 0.021) tended to decreased but did not reach the level of significance. Caregivers’ depression (p = 0.111) and children’s somatic symptoms (p = 0.263) did not change significantly.” Please refer to line 241-247.

“The correlation between affiliate stigma and caregivers’ depression from 0.508 at baseline decrease to 0.407 at follow-up. Although the correlations of affiliate stigma with children’s behavioral problems were not the same compared between at baseline and at follow-up, the correlations did not reach a large and significant level.” Please refer to line 234-239.

Comment 5
The article talks about the difficulty that stigma and blaming of families can pose, and concludes by stressing that it is the parents who need treatment, psycho-educational or cognitive-behavioural, all of which can have an impact on blaming.  I believe that a systemic approach would improve the outlook on this situation and would de-blame the families.

Response

Thank you for your suggestion. We agree that mitigating affiliate stigma needs to be proceeded from individual and environmental levels. We revised the paragraph in section 4.3. Implications as below. Please refer to line 328-339.

“According to Ecological Systems Theory [41], the work of mitigating affiliate stigma should be started form the individual and environmental levels simultaneously. Regarding the individual level, because knowledge is essential to caregivers’ positive attitudes toward children and themselves, behavioral training programs for caregivers should increase their knowledge regarding the etiologies, illness courses, and treatment models of ADHD [42,43]. Training programs should also increase caregivers’ resistance to other people’s negative attitudes [29,44]. Regarding the environmental level, enhancing social support for caregivers may increase their positive attitudes toward themselves and reduce social isolation. Moreover, mitigating public stigma toward ADHD is also essential to decreasing caregivers’ affiliate stigma.”

Round 2

Reviewer 3 Report

I believe that the comments and suggestions raised have been adequately addressed. The changes made help to better understand the research and its implications.